# Agricultural products: A study on the impact of social presence on customer engagement in livestream marketing

**Qiheng Sun[1], Jinying Cao[2]\*, Chen Guo[3], Yuanhao Shi[4]**

1 School of Business, Qinghai Institute of Technology, Xining, Qinghai, China, 2 Business administration, Bangkok Thonburi University, Bangkok, Thailand, 3 School of Economics and Management, Yingkou Institute of Technology, Yingkou, Liaoning, China, 4 School of Economics and Management,Qinghai Minzu University, Xining, Qinghai, China

\* caojy@qhit.edu.cn

## Abstract

With the development of the digital economy, livestream marketing for agricultural products has become an important channel for promoting the digital transformation of rural industries. Drawing on user value theory, this study examines how social presence influences customer engagement in agricultural livestream marketing by incorporating customer satisfaction as a mediating variable and trust atmosphere as a moderating variable, and proposes a conceptual model. Based on data from 312 valid questionnaires, the results show that social presence positively affects customer engagement both directly and indirectly through customer satisfaction. Moreover, trust atmosphere strengthens the direct effect of social presence on customer engagement as well as the mediating role of customer satisfaction. This study extends consumer behavior research by integrating livestream marketing into the theoretical framework and highlighting the role of livestream-specific relational factors in shaping customer behavior. The findings also provide actionable insights for optimizing agricultural livestream marketing, including fostering trust, enhancing interactivity, and supporting farmers and e-commerce platforms.

## 1. Introduction

In the past few years, the global e-commerce market has maintained rapid growth, with the emergence of a variety of online marketing methods, profoundly changing consumer shopping habits and the market landscape. Livestreaming marketing, as an innovative, real-time interactive marketing model, has achieved remarkable development across multiple industries, particularly in the agricultural products sector. In China, in particular, the livestreaming e-commerce market has seen rapid growth in recent years. According to the latest industry data, as shown in Fig 1, China's total livestreaming e-commerce transaction volume is projected to reach 5.1 trillion RMB

**Data availability statement:** All relevant data are within the manuscript and its Supporting information files.

**Funding:** This research was funded by "Kunlun Talent"Introduction Research Project of Qinghai Institute of Technology, grant number 2023-QLGKLYCZX-030. The funder provided access to databases and electronic library resources, as well as financial support for the conduct of the study. The funder had no role in the study design, data collection and analysis, decision to publish, or preparation of the manuscript.

**Competing interests:** The authors have declared that no competing interests exist.

in 2024, a significant year-on-year increase. The sales share of agricultural products continues to increase, demonstrating a high level of consumer recognition and growing demand for livestreaming marketing for agricultural products.

The agricultural products sector presents significant unique challenges. Traditional sales channels often face long supply chains, low efficiency, and information opacity. Livestreaming marketing, with its highly interactive, immediacy, and social nature, effectively overcomes these challenges. Specifically, through livestreaming marketing, consumers can interact with merchants in real time, gaining intuitive insights into key information such as the origin, quality, and production process of agricultural products, significantly enhancing consumer trust and purchasing intent. Furthermore, livestreaming not only expands the agricultural product market but also provides farmers and small agricultural enterprises with new sales channels, helping them increase income and enhance brand awareness, thereby reshaping the traditional agricultural market landscape. As shown in Fig 2, an increasing number of consumers report increased trust in the products and a greater propensity to purchase after watching livestreaming agricultural product displays. This phenomenon demonstrates the powerful consumer influence of livestreaming marketing.

Given the crucial role of livestreaming in influencing consumer purchasing decisions, this study aims to broaden the scope of consumer behavior research by exploring how social presence, customer satisfaction, and a climate of trust interact to influence customer engagement behavior. Through this research, we hope to provide new theoretical perspectives and practical guidance for this field.

Traditional consumer behavior theories primarily focus on purchase decisions, brand loyalty, perceived value, and attitude change. The classic AIDA model (Attention, Interest, Desire, Action) describes the linear process from initial product awareness to final purchase [1,2]. However, this model fails to fully consider the impact of interactivity and social interaction on customer decision-making in livestream marketing. In agricultural product livestream marketing, customers rely not only on product information but are also significantly influenced by the immediate interaction between the host and the audience. This socially driven purchasing decision-making highlights the limitations of traditional models.

In brand loyalty research, traditional theories posit that customer loyalty is formed through continued purchases and accumulated experience [3,4]. However, with the development of social platforms and livestream marketing, brand loyalty increasingly relies on social interaction and emotional connections between customers and brands [5,6]. In particular, in livestream marketing for agricultural products, customers develop a sense of brand identification through interactions with the host and other viewers, which in turn influences their brand loyalty. This phenomenon has been underexplored in traditional brand loyalty theory.

Perceived value theory emphasizes how customers evaluate the functional and emotional value of a product [7,8]. However, in livestream marketing, customers' perceived value comes not only from the product itself but also from the emotional satisfaction and social value gained through interaction. In livestream marketing of

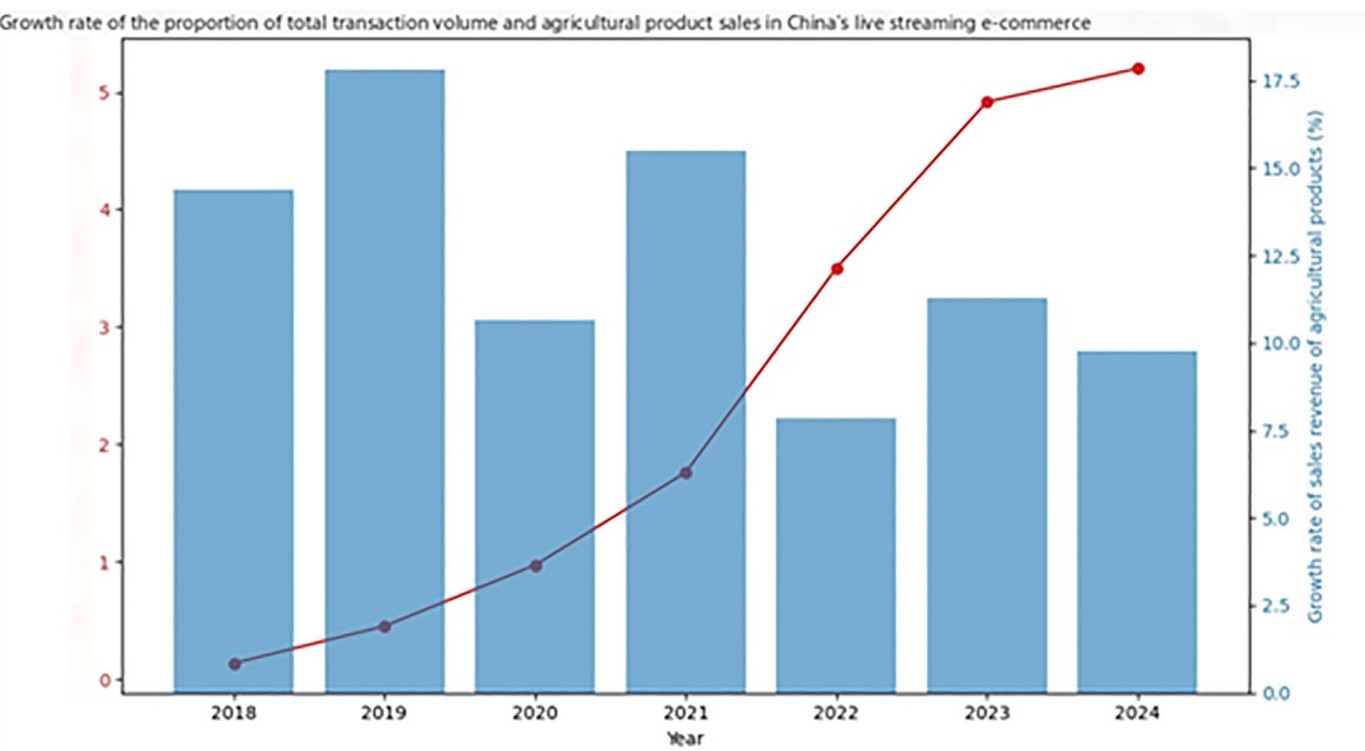

**Fig 1. Growth Rate of Total Livestreaming E-commerce Transaction Volume and Agricultural Products Sales Share in China.**

agricultural products, customer perceived value is profoundly influenced by the interactive process and emotional connection, factors that traditional perceived value theory fails to fully capture.

Regarding attitude change, traditional theories posit that customer attitude change is gradual and stable [9]. However, in livestream marketing, customer attitudes can rapidly shift due to real-time interactions, particularly influenced by social feedback and the host's performance. This shift is particularly pronounced in livestream marketing for agricultural products, where customer attitudes are significantly influenced by social interaction and emotional connection, contradicting the static assumptions of traditional attitude change models.

In summary, traditional consumer behavior theories provide a solid foundation for understanding customer purchasing decisions and brand loyalty, but they expose significant limitations in the context of modern digital marketing, particularly livestream marketing. By enhancing social interaction and emotional engagement, livestream marketing alters the traditional understanding of how customer decision-making and brand loyalty are formed. Against this backdrop, while existing literature has explored the impact of social media and e-commerce on consumer behavior [10–12], the precise mechanisms by which social presence influences customer engagement in livestream marketing have not yet been definitively established. Specifically, the mediating role of customer satisfaction and the moderating effect of a trust climate in highly interactive livestream environments remain insufficiently understood. To address these research gaps, this study integrates social presence and trust climate into a comprehensive consumer behavior framework. A serial mediation model is proposed to elucidate how livestream marketing of agricultural products transforms product information into a credible and empathetic experience through enhanced interaction and trust, thereby extending the theoretical boundaries of conventional consumer behavior theories within livestream contexts.

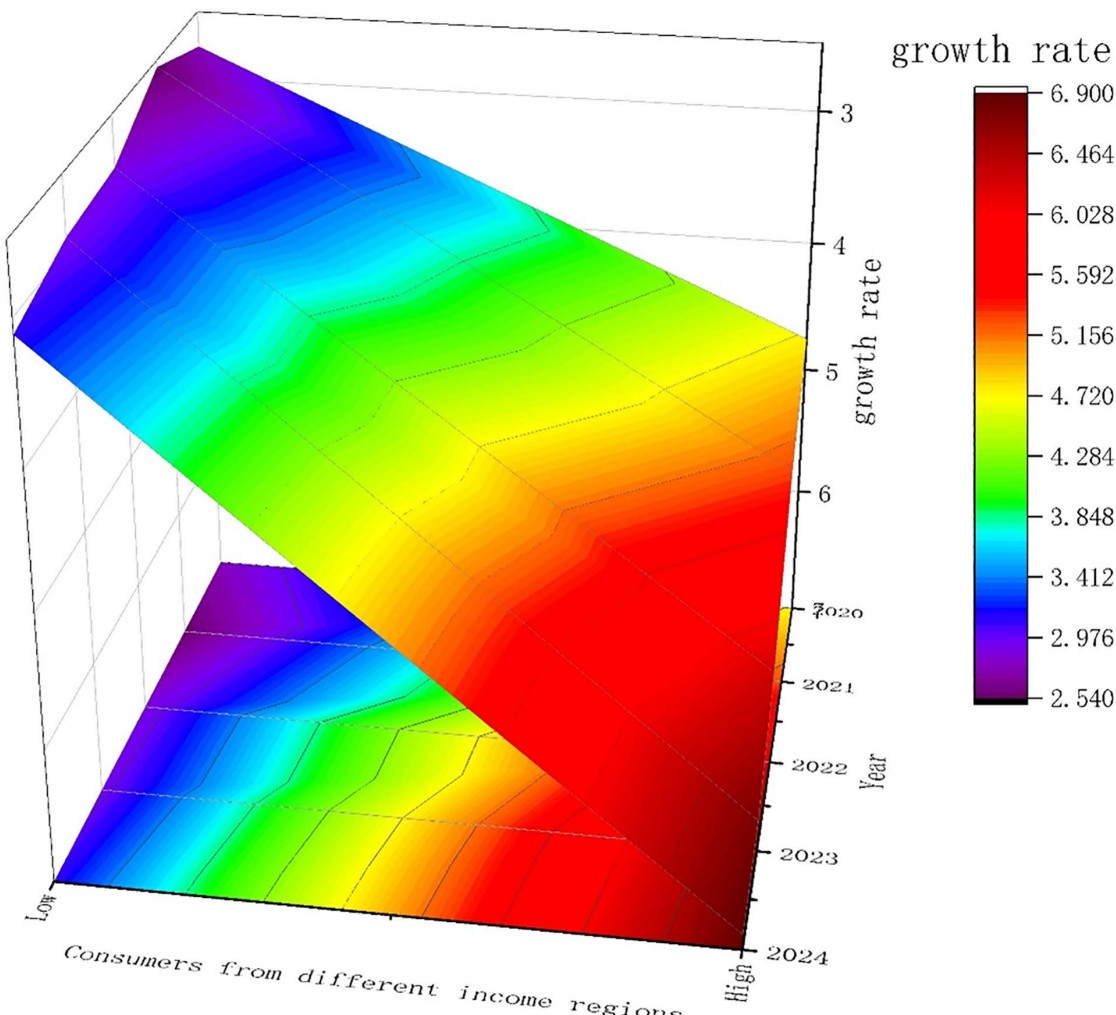

**Fig 2. Growth Rate of Consumer Purchase Intentions Induced by Livestreaming Agricultural Product Displays.**

This study proposes an innovative integrative theoretical framework that incorporates social presence, customer satisfaction, and trust climate. The framework not only overcomes the explanatory limitations of traditional consumer behavior theories in agricultural livestreaming contexts but also provides a novel perspective that is both theoretically grounded and practically oriented for optimizing livestream marketing strategies for agricultural products.

Therefore, the marginal contributions of this paper are as follows:

First, expanding traditional consumer behavior theory: This paper expands traditional consumer behavior theory by introducing the emerging marketing model of livestreaming agricultural product displays. By considering factors such as social presence, customer satisfaction, and a climate of trust, this paper updates existing theories, making them more adaptable to the modern digital marketing environment, particularly in the agricultural products sector.

Second, Revealing Interactive Factors in Agricultural Products Livestream Marketing: This paper systematically explores the interactive effects of factors such as social presence, customer satisfaction, and a climate of trust in agricultural products livestream marketing, filling a gap in the existing literature on how livestream marketing influences customer

                                                                 

engagement behavior in the agricultural products sector and providing a new theoretical framework for related research.

Third, Integrating Theory and Practice: Drawing on the practical application of livestream marketing for agricultural products, this paper provides practical advice for academics and marketing practitioners, offering an important reference for optimizing agricultural products marketing strategies.

## 2. Literature review and research hypotheses

This section reviews the relevant literature on social presence, customer satisfaction, trust climate, and customer engagement within the context of digital and livestream marketing. By synthesizing prior theoretical and empirical studies, this study identifies key research gaps and formulates a set of research hypotheses to elucidate the mechanisms through which social presence influences customer engagement in agricultural livestream marketing.

### 2.1 Social presence and customer engagement behavior

Customer engagement behavior, characterized by both transactional behaviors (such as purchase and repeat consumption) and non-transactional behaviors (such as word-of-mouth recommendations and online interactions) [13], is becoming a hot topic in the field of livestream marketing in the digital economy era. Zhang and Xu [14] emphasizing interaction as the core of value co-creation, this perspective highlights that real-time interactions between consumers, brands, and streamers during live streaming enable consumers to access more product information, enhance interaction value, and foster emotional resonance, thereby promoting consumer engagement and driving the development of social commerce. Huang et al. [15] believe that the sense of social presence consumers experience during livestreaming marketing is a key factor influencing consumer behavior. Ming et al. [16] through a survey and analysis of consumers engaging in online livestreaming shopping, found that enhancing social presence helps build trust between buyers and sellers. Social presence plays a positive role in customer engagement in e-commerce livestreaming scenarios. By enhancing customers' sense of social presence, livestreaming platforms can help stimulate their self-efficacy, thereby increasing their willingness to purchase during livestreaming marketing. Sun et al. [17] through a statistical analysis of e-commerce livestreaming big data, believe that enhancing customers' sense of social presence during livestreaming marketing can significantly improve customer stickiness, thereby helping to increase consumers' willingness to purchase. Existing research shows that the role of "customer engagement" (CE) in cocreating customer experience and value is receiving increasing attention from business practitioners and academics alike [18]. Relying on digital scenarios to enhance social presence during agricultural product livestreaming has become a primary method for enhancing customer engagement.

In agricultural product livestream marketing, social presence plays a more critical role than in many other industries. Agricultural products are often characterized by substantial quality variability and opaque production processes, which can readily result in information asymmetry between consumers and suppliers. Livestream marketing mitigates this issue by creating a strong sense of social presence through real-time visual displays, on-site tasting demonstrations, and original location scene reconstruction. These features not only effectively reduce information asymmetry but also foster an immersive experience that enhances consumers' trust and emotional connection, thereby significantly increasing their engagement intentions and purchase propensity.

Therefore, regarding agricultural product livestreaming marketing, this article proposes the following hypotheses:

H$_1$: Social presence in livestream marketing positively impacts customer engagement.

### 2.2 The mediating role of customer satisfaction

Academics generally believe that customers experience satisfaction when their consumption experience exceeds their expectations. According to the value-satisfaction causal theory, customer satisfaction can significantly and positively

influence customer behavior [19]. In the context of agricultural product e-commerce livestreaming, the customer experience is jointly created by the company and the customer. Digital technology enables customer participation and interaction, further enhancing customer value perception [17]. As a key factor influencing customer behavior, customer satisfaction has long been a key focus of marketing academia and industry. In their study of brand value, Brakus et al. [20] found that customer satisfaction mediates the relationship between experiential value and purchase intention, indicating that customers transition from experience to satisfaction and ultimately to product purchase behavior. Viswanathan et al. [21] believe that customer engagement behavior in the mobile internet environment is a deeper level of behavior that arises when customer satisfaction reaches a certain level. L. Shen et al. [22] found that e-commerce platforms can significantly increase customer stickiness and stimulate customer engagement behavior by improving customer satisfaction.

Within the context of agricultural product livestreaming as a marketing channel, customer satisfaction can be enhanced through detailed presentations of production processes, real-time responsiveness from hosts, and attentive service. For instance, in a livestream promoting "freshly picked high-mountain strawberries" hosted by a local young entrepreneur, visual evidence of product quality and immediate explanations were provided in response to viewers' real-time inquiries, while handwritten thank-you cards were included with orders to convey emotional commitment beyond transactional exchanges. When consumers perceive that their expectations regarding product information, service responsiveness, and emotional interaction are adequately fulfilled, their overall satisfaction increases significantly. This heightened level of satisfaction, in turn, fosters deeper customer engagement, manifested not only in purchase behavior but also in sustained participatory activities such as interacting during livestream sessions, recommending products to others, and maintaining long-term followership of the channel.

Therefore, this article proposes the following hypothesis:

$H_2$: Customer satisfaction mediates the relationship between social presence in livestream marketing and customer engagement.

## 2.3  The moderating role of trust climate

Customer trust is a crucial prerequisite in the field of marketing and is key to customers' choice of e-commerce platforms and acceptance of agricultural product information in live-streaming marketing of agricultural products [23]. Wongkitrungrueng & Assarut [24] founded the role of live streaming as a direct selling tool that has potential to build customer engagement. In other words, unlike traditional agricultural product marketing models, live-streaming marketing of agricultural products connects e-commerce platforms, live-streamers, farmers, and customers. Customers must not only establish trust with the agricultural product sellers but also with the e-commerce platform, live-streamers, and farmers simultaneously, thus forming a trust climate in live-streaming marketing.

In livestream marketing practice, although some agricultural product hosts may attract attention through short-term traffic, their distribution channels are often difficult to sustain in the long term without a consistently high level of trust. Establishing and maintaining a trust climate systematically alleviates consumers' concerns regarding product authenticity and post-purchase risks, thereby providing critical situational support for the transformation of social presence into deep customer engagement behaviors. Specifically, a favorable trust climate not only facilitates the accumulation of loyal customers and the formation of positive word-of-mouth but also enhances the psychological effectiveness of social presence during interactive exchanges. When consumers are embedded in a high-trust climate, they tend to interpret social interaction cues—such as real-time responsiveness from the host and group interaction dynamics—more positively, which in turn more readily stimulates engagement behaviors, including active participation, content co-creation, and long-term relational investment.

Therefore, this article proposes the following hypothesis:

H$_3$: Trust climate strengthens the relationship between social presence and customer engagement. Specifically, the positive correlation between social presence and customer engagement is more pronounced when the trust climate is high than when it is low.

Given that product characteristics such as the freshness and taste of agricultural products typically require actual consumer experience to truly grasp, the delayed nature of the agricultural product experience presents consumers with risks such as mismatched goods and a poor experience in live agricultural product marketing. Establishing a positive atmosphere of trust is a key factor in mitigating these risks in online agricultural product consumption [25]. Wongkitrungrueng & Assarut [24] believe that trust typically mediates the relationship between customer experience perception and behavior in online marketing. Live agricultural product marketing environments with a greater atmosphere of trust are more conducive to enhancing customer engagement, knowledge sharing, and purchasing behaviors.

When consumers adopt a defensive mindset, they are more likely to interpret hosts' interactive behaviors as strategic sales tactics or performative acts, which can prolong purchase deliberation and impede the effective translation of social presence into genuine customer satisfaction. In contrast, when consumers perceive a high-trust climate within agricultural product livestream environments, their risk-avoidance tendencies and cognitive defense mechanisms are substantially attenuated. Under such conditions, customers are more inclined to participate in livestream interactions with an open and engaged orientation. Consequently, the positive experiences generated by social presence can be more fully absorbed and internalized within an unencumbered psychological state, thereby significantly enhancing customer satisfaction. This trust-based satisfaction is not only more enduring but also serves as a robust and stable intrinsic driver of subsequent customer engagement behaviors.

Therefore, this paper proposes the following hypothesis:

H$_4$: Trust climate moderates the mediating effect of customer satisfaction on the impact of social presence on customer engagement behavior. Specifically, the mediating effect of customer satisfaction is stronger when the trust climate level in livestream marketing is high than when it is low.

In summary, the proposed hypotheses advance consumer behavior theory by extending traditional engagement models into the context of livestream marketing. By incorporating social presence as a core experiential factor, and trust climate and customer satisfaction as key contextual moderators, this study offers a more nuanced explanation of how customer engagement is formed in digital and interactive marketing environments, with a specific focus on the agricultural product sector. The research model proposed in this paper is shown in Fig 3.

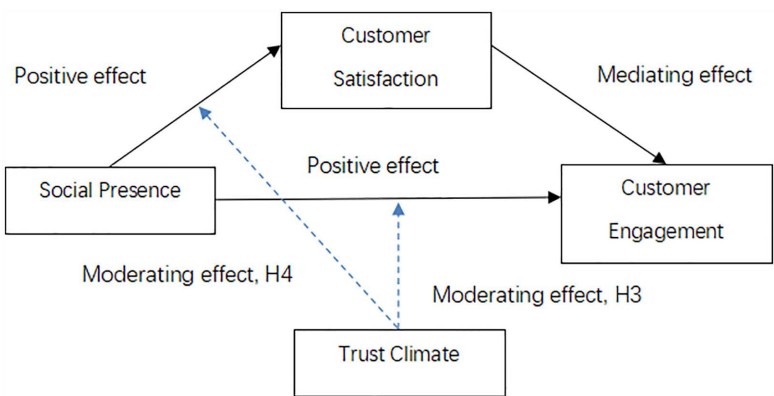

**Fig 3. Research Model.**

## 3. Research design

### 3.1 Sample and data

Data for this study were collected via the Wenjuanxing survey platform using a snowball sampling method, this paper distributed 350 electronic questionnaires to consumers with experience in live-streaming agricultural product marketing. To ensure data quality, 38 invalid questionnaires were excluded due to duplicate completion, too short a time, or too long a time to complete, 312 valid questionnaires were collected, resulting in an effective response rate of 89.1%. Data collection was conducted from April to June 2025, spanning approximately two months. Among the valid questionnaires, 180 respondents were female (57.7%); 235 respondents were aged 18–40 (75.3%); 178 respondents had a bachelor's degree (57.1%); and 228 respondents had a monthly income of 5,000 yuan or less (73.1%). The survey respondents were disproportionately female and young, reflecting the demographic characteristics of livestream marketing, indicating that the sample data for this study generally aligns with survey expectations.

This study was conducted in accordance with the principles of the Declaration of Helsinki. Ethical approval for this research was granted by the School of Business, Qinghai Institute of Technology.

This questionnaire-based study was conducted in accordance with ethical principles for research involving human participants. Participation was voluntary, and written informed consent was obtained from all participants prior to participation.

### 3.2 Measurement of variables

The measurement items for social presence, customer satisfaction, trust atmosphere, and customer engagement behavior used in this study were all based on established, well-established scales recognized by the academic community. Furthermore, based on expert advice and the Chinese context of livestream agricultural product marketing, semantic revisions were made to the wording of individual items. All measurement items were measured using a 7-point Likert scale.

Independent Variable: The measurement of social presence was primarily based on the research of Lu et al. [26], and combined with feedback from experts and scholars during the pilot survey. The final three measurement items were: "When watching livestream marketing, I can feel the enthusiasm of others," "When watching livestream marketing, other people's emotions can also affect me," and "When watching livestream marketing, I can feel the presence of others."

Dependent variable: The measurement of customer engagement behavior mainly draws on the research of Brodie et al. [18], and combines the feedback and suggestions of experts and scholars during the preliminary survey to finally determine the four measurement items: "I am willing to purchase things through live marketing media", "I am willing to continue to participate in live marketing activities in the future", "When watching live marketing, I will refer to other people's suggestions", and "When watching live marketing, I am willing to share and promote my experience".

Mediating Variable: The measurement of customer satisfaction was primarily based on research of Rew et al. [27]. In combination with feedback from experts and scholars during the pilot survey, three measurement items were ultimately selected: "Participating in livestream marketing meets my psychological expectations," "The products purchased through livestream marketing meet my expectations," and "I am satisfied with livestream marketing overall."

Moderator Variable: The measurement of trust climate draws heavily on research of Aslam et al. [28]. In combination with feedback from experts and scholars during the pilot survey, we ultimately determined three measurement items: "I feel that the livestream marketing medium is trustworthy," "I feel that the merchants in livestream marketing are trustworthy," and "I feel that the information provided by livestream marketing is reliable."

## 4. Empirical analysis

### 4.1 Reliability and validity tests

This paper used SPSS statistical software to conduct a reliability analysis of variables such as social presence, customer satisfaction, trust climate, and customer engagement behavior. The results of the composite reliability and convergent

validity tests for the key constructs are presented in Table 1. The measurement scales for all variables exhibit satisfactory internal consistency, convergent validity, and discriminant validity, and the overall model fit meets acceptable thresholds. The Cronbach's alpha coefficient for the total scale was 0.888, the alpha coefficient for the social presence scale was 0.902, the alpha coefficient for the customer satisfaction scale was 0.881, the alpha coefficient for the trust climate scale was 0.849, and the alpha coefficient for the customer engagement scale was 0.905. The alpha coefficients for all scales were greater than the measurement standard of 0.7, indicating that the scales have a high level of reliability. At the same time, using the factor loadings of the measurement items for the social presence, customer satisfaction, trust climate, and customer engagement behavior variables, the combined reliability (CR) of each subscale variable was calculated to be 0.904, 0.885, 0.857, and 0.911, respectively. The combined reliability (CR) values were greater than the measurement standard of 0.6, indicating that the reliability of the questionnaire was ideal.

This paper used AMOS statistical software to conduct a confirmatory factor analysis on the variables of social presence, customer satisfaction, trust climate, and customer engagement behavior. The results showed a CMIN DF value of 2.799, a CFI value of 0.964, a GFI value of 0.930, a TLI value of 0.952, an IFI value of 0.964, and an RMSEA value of 0.076. All fit indices met the test criteria, indicating a good fit between the variables of social presence, customer satisfaction, trust climate, and customer engagement behavior. Furthermore, as shown in the convergent validity test results in Table 1, the factor loadings for social presence, customer satisfaction, trust climate, and customer engagement behavior were all between 0.5 and 0.95, indicating that the survey scale has good convergent validity. The average variance extracted (AVE) for all variables was greater than 0.6, and the correlation coefficients of each variable with other variables were less than the square root of the average variance extracted (AVE), indicating that the discriminant validity of this questionnaire reached an ideal level.

## 4.2 Common method bias tests

This paper used SPSS statistical software to conduct an exploratory factor analysis on the questionnaire items covering social presence, customer satisfaction, trust climate, and customer engagement behavior. The first common factor explained 46.1% of the total variance, meeting the test criteria, indicating that our survey sample was free of significant common method bias. Furthermore, the correlation coefficients between social presence, customer satisfaction, trust climate, and customer engagement ranged from 0.719 to 0.852, with all correlation coefficients below the 0.9 test criteria, further confirming that our survey was minimally affected by homology bias.

**Table 1. Combined Reliability and Convergent Validity Tests.**

| Variables | Measurement Items | Factor Loadings | alpha coefficient | CR | AVE |
|---|---|---|---|---|---|
| Social Presence | SP1 | 0.904 | 0.902 | 0.904 | 0.760 |
| | SP2 | 0.817 | | | |
| | SP3 | 0.891 | | | |
| Customer Satisfaction | CS1 | 0.909 | 0.881 | 0.885 | 0.720 |
| | CS2 | 0.845 | | | |
| | CS3 | 0.788 | | | |
| Trust Climate | TC1 | 0.799 | 0.849 | 0.857 | 0.670 |
| | TC2 | 0.942 | | | |
| | TC3 | 0.696 | | | |
| Customer Conformity Behavior | CEB1 | 0.835 | 0.905 | 0.911 | 0.723 |
| | CEB2 | 0.661 | | | |
| | CEB3 | 0.924 | | | |
| | CEB4 | 0.950 | | | |

## 4.3 Hypothesis testing

This study used SPSS statistical software to conduct hierarchical regression analysis of the research model. Table 2 presents the results of the hierarchical regression analysis, showing that social presence has a significant positive effect on both customer engagement and customer satisfaction, while trust climate exerts a significant moderating effect within this relationship. Specifically, The results showed that in Model 1, the regression coefficient for social presence was significant ($\beta = 0.803$, $p < 0.001$), indicating that social presence in livestream marketing significantly and positively impacted customer engagement behavior, confirming Hypothesis $H_1$. The significant impact of social presence on customer engagement behavior aligns with the findings of Huang et al. [15], which highlight social presence as a critical driver of consumer engagement in live commerce environments. In Model 2, the regression coefficients for social presence ($\beta = 0.565$, $p < 0.001$) and customer satisfaction ($\beta = 0.326$, $p < 0.001$) were significant, with social presence being less significant in Model 2 than in Model 1. In Model 3, the regression coefficient for social presence was significant ($\beta = 0.731$, $p < 0.001$), indicating that social presence in livestream marketing had a significant positive impact on customer satisfaction. In summary, customer satisfaction partially mediates the relationship between social presence in livestream marketing and customer engagement behavior.

To fully verify the mediating effect of customer satisfaction, this paper used SPSS Process to conduct a bias-corrected nonparametric percentile bootstrap interval test with a sample size of 5000 and a confidence level of 95%. The test results are shown in Table 3: The mediation effect value of the bootstrap mediation test is 0.238, and the 95% confidence interval is [0.161, 0.322], which does not include 0. This indicates that customer satisfaction mediates the relationship between social presence in livestream marketing and customer engagement behavior, and the mediating effect of customer satisfaction is significant. Accordingly, the results reported in Table 3 indicate that social presence not only exerts a direct effect

**Table 2. Hierarchical Regression Test Results.**

| Model | Customer Conformity Behavior | | Customer Satisfaction | |
|---|---|---|---|---|
| | Model 1 | Model 2 | Model 3 | Model 4 |
| Gender | −0.143* (−1.951) | −0.099 (−1.471) | −0.136 (−1.463) | −0.125 (−1.550) |
| Age | 0.019 (0.539) | 0.012 (0.371) | 0.021 (0.485) | 0.020 (0.535) |
| Education | 0.018 (0.326) | 0.016 (0.324) | 0.005 (0.074) | −0.021 (−0.340) |
| Occupation | 0.001 (0.020) | −0.002 (−0.127) | 0.006 (0.232) | 0.014 (0.632) |
| Income | 0.004 (0.173) | 0.016 (0.681) | −0.036 (−1.089) | −0.035 (−1.235) |
| Social Presence | 0.803*** (25.123) | 0.565*** (13.480) | 0.731*** (18.038) | 0.361*** (6.818) |
| Customer Satisfaction | | 0.326*** (7.894) | | |
| Trust Climate | | | | 0.528*** (9.969) |
| Social Presence × Trust Climate | | | | 0.094*** (3.387) |
| Adjusted R$^2$ | 0.673 | 0.728 | 0.513 | 0.642 |

Note: *** indicates $P < 0.001$, ** indicates $P < 0.01$, * indicates $P < 0.05$, and the t value is in brackets.

**Table 3. Bootstrap Test Results for Mediation.**

| Effect Category | Effect Coefficient | Standard Error | 95% Confidence Interval | |
|---|---|---|---|---|
| | | | Lower Bound | Upper Bound |
| Direct Effect | 0.565 | 0.042 | 0.483 | 0.648 |
| Indirect Effect | 0.238 | 0.041 | 0.161 | 0.322 |

on customer engagement but also influences customer engagement indirectly through customer satisfaction, thereby providing evidence of partial mediation. At this point, hypothesis H2 has been verified again. The mediating role of customer satisfaction supports the value-satisfaction causal theory proposed by Woodruff ([19]), which suggests that satisfaction is a key driver of customer behavior.

This paper uses SPSS statistical software to separately center the independent variable, social presence, and the moderating variable, trust climate, in the moderation model. The interaction term is generated by multiplying the centered variables to ensure that the coefficients are more explanatory. As shown in Table 2, Model 4 is the test equation for the moderating effect of Hypothesis H3. The independent variables are social presence, trust climate, and social presence×trust climate, and the dependent variable is customer satisfaction. The results show that the coefficient for the interaction term, social presence×trust climate, is significant ($\beta=0.094$, $p<0.001$), indicating that the moderating effect of trust climate is significant, confirming Hypothesis H3. This finding is consistent with trust theory [23], which emphasizes trust climate as a ffundamental basis for relationship development and behavioral commitment, particularly in environments characterized by high uncertainty.

This paper uses the SPSS Process plug-in to test the moderated mediation effect. As can be seen from the left panel of Table 4, presents the moderated mediation results, indicating that the indirect effect of social presence on customer engagement via customer satisfaction is stronger under a high-trust climate, thereby underscoring the contextual importance of trust in agricultural livestream marketing. When the trust climate is 1 standard deviation below the mean, the indirect effect of social presence in live marketing on customer engagement behavior through customer satisfaction is 0.083 (95% CI does not include 0), indicating a significant moderated indirect effect. When the trust climate is 1 standard deviation above the mean, the indirect effect of social presence in live marketing on customer engagement behavior through customer satisfaction is 0.152 (95% CI does not include 0), indicating a significant moderated indirect effect. Furthermore, the INDEX value in the right panel is 0.031 (95% CI does not include 0), indicating that trust climate significantly moderates the indirect effect of social presence in live marketing on customer engagement behavior. Specifically, trust climate strengthens the mediating role of customer satisfaction in the relationship between social presence in live marketing and customer engagement behavior, confirming Hypothesis H4. This is especially relevant in agricultural product livestream marketing, where product quality, freshness, and safety are difficult to verify prior to consumption, making trust a critical contextual factor influencing customer engagement decisions.

## 5. Main conclusions and policy implications

### 5.1 Key findings

The digital transformation of rural industries is a key path for China to achieve rural revitalization. Vigorously developing live e-commerce for agricultural products is not only advocated by national policy, but also an important part of building an innovative ecosystem for agricultural products in the digital economy era. Based on user value theory, this paper explores the relationship between social presence and customer engagement in the context of agricultural product e-commerce live streaming. A conceptual model of social presence, customer engagement, customer satisfaction, and a climate of trust is constructed. Through an empirical analysis of 312 valid questionnaires, the following conclusions are drawn:

**Table 4. Bootstrap Test Results for Moderated Mediation.**

| Moderated Indirect Effect | | | | | Moderated Mediation Effect | | |
|---|---|---|---|---|---|---|---|
| Moderator Variable | Effect Coefficient | Standard Error | 95% Confidence Interval | | INDEX | 95% Confidence Interval | |
| | | | Lower Bound | Upper Bound | | Lower Bound | Upper Bound |
| Low Value | 0.083 | 0.029 | 0.030 | 0.143 | 0.031 | 0.006 | 0.057 |
| High Value | 0.152 | 0.034 | 0.091 | 0.226 | | | |

First, social presence in agricultural product e-commerce live streaming significantly promotes customer engagement, with customer satisfaction mediating the relationship. This suggests that agricultural product live streaming, characterized by real-time interaction, immediate feedback, and emotional resonance, provides a higher quality social atmosphere than traditional agricultural product marketing models. In the agricultural product live streaming marketing process, platform hosts leverage digital technology to achieve more direct and authentic interactions with customers, significantly increasing consumer engagement and experience, making it easier for consumers to meet their expectations and thus enhance customer engagement.

Second, in the context of agricultural product e-commerce live streaming, a climate of trust positively moderates the relationship between social presence and customer engagement, as well as the mediating effect of customer satisfaction. This result highlights the importance of building a climate of trust in the live-streaming marketing of agricultural products. Only by establishing trust between e-commerce platforms, farmers, and consumers can the effectiveness of live-streaming marketing be fully enhanced.

## 5.2 Theoretical contributions

Building on the limitations of traditional consumer behavior theories discussed in the Introduction, this study advances consumer behavior theory by demonstrating that customer decision-making in livestream marketing is inherently dynamic and interaction-driven. Prior research has emphasized real-time interaction and emotional resonance as critical determinants of consumer behavior in livestreaming contexts. By empirically validating the direct effect of social presence on customer engagement, this study moves beyond static, transaction-oriented models and highlights the necessity of a dynamic, interaction-focused framework to understand consumer behavior in digital marketing environments, particularly within agricultural product livestreaming.

From a theoretical standpoint, the mediating role of customer satisfaction identified in this study provides robust empirical support for user value theory and the value-satisfaction causal framework. While traditional applications of these theories have predominantly focused on functional and economic value, the findings demonstrate that social presence constitutes a critical antecedent shaping customer satisfaction and subsequent engagement behaviors. This extends the value-satisfaction causal framework by incorporating social presence as an upstream value driver, thereby enhancing its relevance and explanatory power in interactive digital marketing contexts.

This study further contributes to trust theory by conceptualizing trust as a contextual climate rather than solely as a dyadic relationship in agricultural livestream marketing. Unlike standardized products, agricultural products exhibit significant quality variability, rendering trust climate a fundamental prerequisite for customer engagement. The empirical evidence that trust climate moderates both the direct and indirect effects of social presence underscores the situational and contextual nature of trust formation in livestream environments. By embedding trust climate into the consumer behavior framework, this study extends trust theory to better account for engagement mechanisms in high-uncertainty digital marketplaces.

## 5.3 Policy recommendations

Based on a systematic analysis of the underlying mechanisms and empirical findings in agricultural product livestream marketing, the following policy recommendations are proposed and prioritized.

First, promote the development and standardization of a trusting atmosphere. Given the inherent quality variability of agricultural products, priority should be placed on cultivating a trust climate that directly fosters sustained customer engagement. This study found that a trusting atmosphere plays a moderating role in agricultural product livestream marketing, significantly improving customer engagement and customer satisfaction. This suggests that building trust relationships is key to improving the effectiveness of agricultural product e-commerce livestreaming. Policy recommendations

should focus on promoting trust between e-commerce platforms, farmers, and consumers through government and industry associations, and establishing a more transparent and standardized livestream marketing system.

Second, strengthen the interactive mechanisms between live-streaming platforms and customers. Building on a foundation of trust, high-quality real-time interaction constitutes the primary mechanism through which social presence operates effectively in livestream marketing. This study shows that social presence in live-streaming marketing of agricultural products significantly promotes customer engagement behavior, with customer satisfaction playing a significant mediating role in this process. To improve customer engagement, policymakers are advised to encourage agricultural product e-commerce platforms to strengthen their customer interaction mechanisms, particularly through technological means to enhance real-time interaction and emotional resonance between livestreamers and customers.

Third, strengthen training and support for farmers and e-commerce platforms. The effective implementation of a favorable trust climate and interactive mechanisms depends on the presence of competent actors. This study found that customer satisfaction plays a significant role in agricultural product livestreaming, particularly the ability of platform hosts to enhance customer engagement through emotional resonance and interaction. To improve the livestreaming marketing capabilities of farmers and e-commerce platforms, policies should strengthen training and technical support for farmers and platforms. The government can enhance farmers' understanding and operational capabilities of livestreaming marketing by organizing training, establishing professional livestreaming guidance teams, and providing technical equipment support.

Finally, promote personalized and differentiated agricultural product livestream marketing. Market segmentation based on consumer characteristics can be effectively implemented to sustainably enhance the livestream marketing channel only after the first three policy recommendations have been executed. According to the conclusions of this study, the interactivity and social atmosphere in agricultural product livestreaming have a profound impact on customer behavior. This provides new insights for agricultural product livestreaming marketing: to improve customer engagement, agricultural product livestreaming should develop differentiated marketing strategies based on the characteristics of different consumer groups.

### 5.4 Limitations and future research

This study has several limitations. First, the sample size is relatively limited, which may affect the generalizability of the findings. Second, this study used cross-sectional data and was unable to explore long-term changes in causal relationships.

Future research can further validate the generalizability of the findings by expanding the sample size. Furthermore, panel data and causal analysis methods are used to explore the dynamics of social presence, customer satisfaction, and trust, as well as their interrelationships.

## Supporting information

**S1 File. Minimal anonymized dataset.**
(XLS)

**S2 File. Questionnaire.**
(DOCX)

## Author contributions

**Conceptualization:** Jinying Cao.

**Data curation:** Jinying Cao.

**Formal analysis:** Yuanhao Shi.

**Funding acquisition:** Qiheng Sun.

**Investigation:** Jinying Cao.

**Project administration:** Qiheng Sun, Jinying Cao.

**Resources:** Qiheng Sun.

**Software:** Chen Guo.

**Supervision:** Chen Guo, Yuanhao Shi.

**Validation:** Qiheng Sun, Chen Guo.

**Visualization:** Chen Guo, Yuanhao Shi.

**Writing – original draft:** Qiheng Sun.

**Writing – review & editing:** Yuanhao Shi.

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
