## [Decision Letter · Decision Letter 0]

20 Nov 2025

Dear Dr. Shi,

Thank you for submitting your manuscript to PLOS ONE. After careful consideration, we feel that it has merit but does not fully meet PLOS ONE’s publication criteria as it currently stands. Therefore, we invite you to submit a revised version of the manuscript that addresses the points raised during the review process.

We look forward to receiving your revised manuscript.

Kind regards,

Sudarsan Jayasingh, Ph.D

Academic Editor

PLOS ONE

Journal Requirements:

This research was funded by “Kunlun Talent”Introduction Research Project of Qinghai Institute of Technology , grant number 2023-QLGKLYCZX-030.

5. We note that you have indicated that there are restrictions to data sharing for this study. PLOS only allows data to be available upon request if there are legal or ethical restrictions on sharing data publicly. For more information on unacceptable data access restrictions, please see http://journals.plos.org/plosone/s/data-availability#loc-unacceptable-data-access-restrictions.

6. We note that your Data Availability Statement is currently as follows: All relevant data are within the manuscript and its Supporting Information files.

Reviewers' comments:

Reviewer's Responses to Questions

**Comments to the Author**

1. Is the manuscript technically sound, and do the data support the conclusions?

Reviewer #1: Yes

Reviewer #2: Yes

2. Has the statistical analysis been performed appropriately and rigorously?

Reviewer #1: Yes

Reviewer #2: Yes

3. Have the authors made all data underlying the findings in their manuscript fully available?

Reviewer #1: Yes

Reviewer #2: Yes

4. Is the manuscript presented in an intelligible fashion and written in standard English?

Reviewer #1: Yes

Reviewer #2: Yes

Reviewer #1: This manuscript investigates the influence of social presence on customer engagement in livestream agricultural product marketing, focusing on the mediating role of customer satisfaction and the moderating effect of trust climate. The topic is timely and relevant given the rapid shifts toward digital agriculture and livestream commerce. The paper is well-structured, and the theoretical grounding is generally solid. The authors provide clear hypotheses, appropriate methods, and meaningful practical implications. Here are my suggestions:

1. The paper claims to extend consumer behavior theory, but existing work on livestream commerce, social presence, and trust is already substantial. The authors should clarify what is novel relative to recent livestream commerce studies (e.g., SOR frameworks, presence-trust-engagement models).

2. The sample (312 respondents) is reasonable, but the sampling method (Wenjuanxing) suggests a convenience sample. Potential sampling bias should be acknowledged. It will be helpful if the authors could provide additional demographic characteristics (e.g., platform usage frequency, livestream categories used).

3. Measurement items are described but not presented in a full instrument table. Many journals require a complete item list in Appendix.

4. While results support hypotheses, theoretical explanations could be expanded.

I hope the authors find the suggestions helpful. Good luck with the project!

Reviewer #2: Thank you for inviting me to review this pape. Overall, the manuscript is well-written and provides valuable insights into livestream marketing for agricultural products. Please see my attaceh review report for comments and improvement. Addressing these commnets will enhance the clarity, coherence, and practical relevance of the paper.

**Do you want your identity to be public for this peer review?** For information about this choice, including consent withdrawal, please see our Privacy Policy

Reviewer #1: No

Reviewer #2: **Yes:** Mouza M. S. Al Hadhrami

---

## [Author Response · Author response to Decision Letter 1]

21 Jan 2026

Response to Reviewers

Manuscript Title:

Agricultural products: A study on the impact of social presence on customer engagement in livestream marketing

Manuscript ID: PONE-D-25-56485

Journal: PLOS ONE

Dear Editor and Reviewers,

We sincerely thank you and the reviewers for the time and effort devoted to reviewing our manuscript. We are grateful for the insightful and constructive comments, which have greatly helped us to improve the quality, clarity, and rigor of our study.

We have carefully revised the manuscript in response to all comments and suggestions. All changes have been highlighted in the revised manuscript using track changes. Below, we provide a detailed, point-by-point response to each comment. For clarity, reviewers’ comments are reproduced in bold, followed by our responses.

We hope that the revisions adequately address the concerns raised and that the revised manuscript will be suitable for publication in PLOS ONE.

Once again, we sincerely appreciate the editor’s and reviewers’ valuable feedback.

Kind regards,

Qiheng Sun, Jinying Cao, Chen Guo, Yuanhao Shi

Corresponding author: Jinying Cao

E-mail: caojy@qhit.edu.cn

Response to Editor’s Comments

Editor’s Comment 1:

Response:

Thank you very much for this reminder. We have carefully revised the manuscript to ensure full compliance with PLOS ONE’s style and formatting requirements. Specifically, we have reformatted the main text, title page, author affiliations, and file naming according to the official PLOS ONE templates provided by the journal. We confirm that the revised submission now adheres to all relevant PLOS ONE formatting guidelines.

Editor’s Comment 2:

Please include a complete copy of PLOS’ questionnaire on inclusivity in global research in your revised manuscript. Our policy for research in this area aims to improve transparency in the reporting of research performed outside of researchers’ own country or community. The policy applies to researchers who have travelled to a different country to conduct research, research with Indigenous populations or their lands, and research on cultural artefacts. The questionnaire can also be requested at the journal’s discretion for any other submissions, even if these conditions are not met. Please upload a completed version of your questionnaire as Supporting Information when you resubmit your manuscript.

Response:

Thank you very much for this important comment and for drawing our attention to PLOS ONE’s policy on inclusivity in global research. We have carefully completed the PLOS questionnaire on inclusivity in global research and included a full copy in the revised submission. The completed questionnaire has been uploaded as Supporting Information and is also referenced in the manuscript on pages 34–38. We confirm that this revision complies with the journal’s transparency and reporting requirements.

Editor’s Comment 3:

Please provide additional details regarding participant consent. In the ethics statement in the Methods and online submission information, please ensure that you have specified (1) whether consent was informed and (2) what type you obtained (for instance, written or verbal, and if verbal, how it was documented and witnessed). If your study included minors, state whether you obtained consent from parents or guardians. If the need for consent was waived by the ethics committee, please include this information.

Response:

Thank you very much for this important comment regarding ethical considerations and participant consent. In response, we have revised the manuscript to explicitly clarify the nature and type of consent obtained from participants. Specifically, in Section 3.1 Sample and Data (paragraph 2-3), we have added a statement indicating that participation was voluntary and that written informed consent was obtained from all participants prior to participation. This revision can be found in the revised manuscript in Section 3.1.

Editor’s Comment 4:

Thank you for stating the following financial disclosure:

This research was funded by “Kunlun Talent” Introduction Research Project of Qinghai Institute of Technology, grant number 2023-QLGKLYCZX-030.

Response:

Thank you very much for this comment. In accordance with the editor’s request, we have clarified the Role of the Funder statement. Specifically, we have added the amended statement in the Cover Letter (paragraph 5), indicating that the funder provided access to databases and electronic library resources, as well as financial support for the conduct of the study, but had no role in the study design, data collection and analysis, decision to publish, or preparation of the manuscript. We appreciate the editor’s assistance in updating the online submission form accordingly.

Editor’s Comment 5:

We note that you have indicated that there are restrictions to data sharing for this study. PLOS only allows data to be available upon request if there are legal or ethical restrictions on sharing data publicly. Before we proceed with your manuscript, please address the following prompts: a) If there are ethical or legal restrictions on sharing a de-identified data set, please explain them in detail (e.g., data contain potentially identifying or sensitive patient information, data are owned by a third-party organization, etc.) and who has imposed them (e.g., a Research Ethics Committee or Institutional Review Board, etc.). Please also provide contact information for a data access committee, ethics committee, or other institutional body to which data requests may be sent. b) If there are no restrictions, please upload the minimal anonymized data set necessary to replicate your study findings to a stable, public repository and provide us with the relevant URLs, DOIs, or accession numbers.You also have the option of uploading the data as Supporting Information files, but we would recommend depositing data directly to a data repository if possible.

Response:

Thank you very much for this important comment regarding data availability. In response, we have revised our data sharing approach to ensure full compliance with PLOS ONE’ s data availability policy. There are no legal or ethical restrictions on sharing the data underlying this study. Accordingly, we have uploaded the minimal anonymized dataset necessary to replicate the study findings as Supporting Information files in the revised submission. All data have been fully de-identified prior to sharing. We believe this revision adequately addresses the journal’ s data availability requirements.

Editor’s Comment 6:

We note that your Data Availability Statement is currently as follows: All relevant data are within the manuscript and its Supporting Information files.

Please confirm at this time whether or not your submission contains all raw data required to replicate the results of your study. Authors must share the “minimal data set” for their submission. PLOS defines the minimal data set to consist of the data required to replicate all study findings reported in the article, as well as related metadata and methods.

-The values behind the means, standard deviations and other measures reported;

-The values used to build graphs;

-The points extracted from images for analysis.

Response:

Thank you very much for this clarification request. We confirm that the revised submission includes the complete minimal dataset required to replicate all findings reported in the study. Specifically, the Supporting Information files contain the raw questionnaire response data underlying all descriptive statistics presented in the manuscript.

Editor’s Comment 7:

Response:

Thank you for this clarification. The reviewer comments did not include recommendations to cite specific previously published works. Therefore, no additional citations were added in response to this comment.

Response to Reviewer #1

Comment 1:

The paper claims to extend consumer behavior theory, but existing work on livestream commerce, social presence, and trust is already substantial. The authors should clarify what is novel relative to recent livestream commerce studies (e.g., SOR frameworks, presence-trust-engagement models).

Response:

We sincerely thank the reviewer for this important and constructive comment. We fully agree that prior research on livestream commerce, social presence, and trust—particularly studies based on SOR frameworks and presence–trust–engagement models—has established a solid theoretical foundation. To more clearly articulate the novelty of our study relative to this existing literature, we have revised the Introduction (paragraph 9) to explicitly highlight our Novelty.

Comment 2:

The sample (312 respondents) is reasonable, but the sampling method (Wenjuanxing) suggests a convenience sample. Potential sampling bias should be acknowledged. It will be helpful if the authors could provide additional demographic characteristics (e.g., platform usage frequency, livestream categories used).

Response:

We sincerely thank the reviewer for this insightful comment. We would like to clarify that the study used a snowball sampling method rather than a simple convenience sample. Specifically, data were collected via the Wenjuanxing survey platform using snowball sampling, which allowed us to reach a sufficiently large and relevant participant pool engaged in agricultural livestreaming.

We have clarified the sampling method in the revised manuscript (Section 3.1 Sample and Data, first paragraph, lines 295–296).

Comment 3:

Measurement items are described but not presented in a full instrument table. Many journals require a complete item list in Appendix.

Response:

Thank you very much for this helpful comment. We have uploaded the full questionnaire and minimal anonymized dataset, as Supporting Information files in the revised submission.

Comment4:

While results support hypotheses, theoretical explanations could be expanded.

Response:

We sincerely thank the reviewer for this valuable suggestion. In response, we have expanded the theoretical explanations for each hypothesis in Section 2 Literature Review and Research Hypotheses. Specifically, we have provided more detailed reasoning and literature support for the relationships between social presence, customer satisfaction, trust climate, and customer engagement in agricultural livestreaming contexts. We believe these revisions strengthen the theoretical grounding of the study and provide a clearer rationale for the proposed hypotheses.

Response to Reviewer #2

Comment 1: The Abstract

Well-structured and decent. My only concern is that it should avoid repetition. The abstract repeats the idea of social presence promoting customer engagement multiple times. Consider streamlining this to avoid redundancy. Also, the practical recommendations are valuable but could be summarized more succinctly. For example, "The study provides actionable insights for optimizing agricultural livestream marketing, including fostering trust, enhancing interactivity, and supporting farmers and e-commerce platforms."

Response:

We sincerely thank the reviewer for this helpful and constructive suggestion. In response, we have streamlined the abstract to reduce redundancy regarding social presence and customer engagement. Additionally, we have incorporated the reviewer’s suggested sentence to succinctly summarize the practical recommendations. The entire abstract has been language-polished and revised for clarity and conciseness. These changes can be found in the revised manuscript on page 2.

Comment 2: Introduction

The introduction section follows a standard academic structure, providing a clear context, identifying gaps in existing research, and outlining the study's objectives and contributions. Main enhancement areas might be:

(1)Lack of a clear research problem statement. Just be more explicit and straightforward, emphasizing that the specific mechanisms by which social presence, customer satisfaction, and trust climate influence customer engagement remain underexplored.

(2)Need to highlight the novelty by emphasizing that the study is among the first to integrate social presence and trust climate into the consumer behaviour framework, providing a fresh perspective on livestream marketing for agricultural products (if I am not mistaken).

(3)The section mentions the limitations of traditional consumer behavior theories, but could explicitly highlight how the study addresses these gaps (paragraph 8)

Response:

(1)Thank you for this valuable suggestion. We agree that the research problem statement needed to be stated more explicitly and directly. In response, we have revised the Introduction to clearly highlight the existing research gap. Specifically, we now emphasize that while social presence has been widely examined in livestream marketing, the specific mechanisms through which it influences customer engagement remain insufficiently understood. Moreover, we explicitly point out the underexplored mediating role of customer satisfaction and the moderating role of trust climate in highly interactive livestream environments.

These revisions have been added in the Introduction (Page 6-7, Paragraph 8, Lines 119–122).

(2)Thank you for this positive and encouraging comment. In response, we have explicitly articulated the theoretical contribution and innovation of this study in the manuscript. Specifically, we have added a clear statement highlighting the development of an integrative theoretical framework that incorporates social presence, customer satisfaction, and trust climate, and emphasized its theoretical and practical significance in the context of agricultural livestream marketing.

These revisions have been added in the Introduction (Page 7, Paragraph 9).

(3)Thank you for this helpful suggestion. We agree that the original text could more clearly explain how the proposed study addresses the limitations of traditional consumer behavior theories. In response, we have revised Paragraph 8 of the Introduction to explicitly describe how the integration of social presence and trust climate within a unified framework addresses these gaps. In particular, we clarify the use of a serial mediation model to explain how livestream marketing transforms product information into a credible and empathetic experience through enhanced interaction and trust, thereby extending the theoretical boundaries of traditional consumer behavior theories in livestream contexts.

These revisions have been added in the Introduction (Page 7, Paragraph 8, Lines 124–128).

Comment 3: Theory and Research Hypotheses

1.I prefer to add “literature review” as a main headline with a brief introduction on how you are about to discuss the Theory and Research Hypotheses. Never mind, if this is done according to the journal template.

2.This part has a clear structure, effectively incorporates prior research to support the hypotheses, which are found clearly and logically derived from the discussion. Here are my points of concern:

(1)The hypotheses are well-stated, but the rationale for each could be expanded slightly in one paragraph under each one. For example:

For H1, explain why social presence is particularly impactful in the agricultural product sector compared to other industries.

For H2, elaborate on how customer satisfaction mediates the relationship between social presence and engagement, perhaps with a brief example or scenario.

For H3 and H4, provide more detail

---

## [Editor Report · Decision Letter 1]

26 Jan 2026

Agricultural products: A study on the impact of social presence on customer engagement in livestream marketing

PONE-D-25-56485R1

Dear Dr. Cao,

We’re pleased to inform you that your manuscript has been judged scientifically suitable for publication and will be formally accepted for publication once it meets all outstanding technical requirements.

Kind regards,

Sudarsan Jayasingh, Ph.D

Academic Editor

PLOS One

Additional Editor Comments (optional):

The authors has addressed all the reviewers comments.

---

## [Editor Report · Acceptance letter]

PONE-D-25-56485R1

PLOS One

Dear Dr. Cao,

I'm pleased to inform you that your manuscript has been deemed suitable for publication in PLOS One. Congratulations! Your manuscript is now being handed over to our production team.

Kind regards,

on behalf of

Dr. Sudarsan Jayasingh

Academic Editor

PLOS One